# COVID-19 Treatment—Current Status, Advances, and Gap

**DOI:** 10.3390/pathogens11101201

**Published:** 2022-10-18

**Authors:** Chian Ho, Ping-Chin Lee

**Affiliations:** 1Faculty of Science and Natural Resources, Universiti Malaysia Sabah, Kota Kinabalu 88400, Sabah, Malaysia; 2Biotechnology Research Institute, Universiti Malaysia Sabah, Kota Kinabalu 88400, Sabah, Malaysia

**Keywords:** COVID-19, peptides, virus life cycle, FDA-approved drugs, drug development

## Abstract

COVID-19, which emerged in December 2019, was declared a global pandemic by the World Health Organization (WHO) in March 2020. The disease was caused by severe acute respiratory syndrome coronavirus 2 (SARS-CoV-2). It has caused millions of deaths worldwide and caused social and economic disruption. While clinical trials on therapeutic drugs are going on in an Accelerating COVID-19 Therapeutic Interventions and Vaccines (ACTIV) public–private partnership collaboration, current therapeutic approaches and options to counter COVID-19 remain few. Therapeutic drugs include the FDA-approved antiviral drugs, Remdesivir, and an immune modulator, Baricitinib. Hence, therapeutic approaches and alternatives for COVID-19 treatment need to be broadened. This paper discusses efforts in approaches to find treatment for COVID-19, such as inhibiting viral entry and disrupting the virus life cycle, and highlights the gap that needs to be filled in these approaches.

## 1. Introduction

COVID-19 emerged in December 2019 and was declared a global pandemic by the World Health Organization (WHO) in March 2020 [1]. As of 15 May 2022, over 518 million confirmed COVID-19 cases and over 6 million deaths have been reported worldwide and have caused devastating social and economic disruption [2]. COVID-19 was caused by severe acute respiratory syndrome coronavirus 2 (SARS-CoV-2), a novel enveloped positive-sense RNA *Betacoronavirus* about 60–140 nm in diameter [3]. The viral genome encodes four structural proteins. These include the spike protein (S), which is necessary for viral entry; the envelope protein (E), which is essential for viral assembly; the membrane protein (M), required for viral morphogenesis; and the nucleocapsid protein (N), which is linked to the viral genome [1,4]. In addition, the viral genome codes for sixteen non-structural proteins (nsp), nine accessory proteins, and antagonists of the host inflammatory response. These proteins are crucial for viral protein translation and genome replication [5,6].

Infection of SARS-CoV-2 begins with virus entry either via viral membrane fusion with the host cell membrane or via endosomes [7]. The S protein possesses the receptor-binding domain (RBD) that attaches to the host cell’s angiotensin-converting enzyme 2 (ACE2) receptor. After the initial attachment, the S protein is cleaved into S1 and S2 subunits by host proteases, such as transmembrane protease serine 2 (TMPRSS2) and furin at the S1/S2 cleavage site. The cleaving of the S protein causes a conformational change that facilitates the fusion of viral and host cell membranes and the subsequent viral genome released into the host cell [8]. However, entry via endosomes has the host protease Cathepsin L and Cathepsin B, cleaving the S protein at the S1/S2 cleavage site at an acidic pH. Fusing the viral membrane with endosomes releases the viral genome into the host cell cytoplasm [9].

Once inside the host cell cytoplasm, the viral RNA is translated into two polyproteins, pp1a and pp1ab, which are subsequently processed into the 16 non-structural proteins. Among these nsps, nsp3, also known as papain-like protease (PLpro), and nsp5, a chymotrypsin-like protease, also known as 3C-like protease (3CLpro), are given particular attention by researchers because of their involvement in post-translational cleaving of the polyproteins.

Several of the nsps, especially nsp12, the RNA-dependent-RNA polymerase (RdRp), form the replication and transcription complex (RTC), which drives the replication of the RNA genome and translation of structural and accessory proteins. Viral assembly occurs and new viruses are released from the host cell via exocytosis and continue to infect other host cells [10].

## 2. Current Approaches to Finding COVID-19 Treatment

Treatment for COVID-19 is designed using the information of the SARS-CoV-2 life cycle. The therapeutic approaches include (i) boosting the host immune system with monoclonal antibodies synthesised in the laboratory that neutralises the virus, (ii) preventing viral entry into host cells by targeting the various proteins involved, (iii) inhibiting the formation of RTC by inhibiting the main protease, (iv) preventing virus genome replication by inhibiting RdRp, and (v) prescribing anti-inflammatory medications and immune modulators to reduce COVID-19 severity and mortality due to host’s hyper inflammation response [11,12].

### 2.1. FDA-Approved and EUA-Authorised Drugs

The disease manifests as mild to moderate respiratory symptoms such as cough, sore throat, and/or fever. Asymptomatic individuals have also been identified, which has added challenges to disease control and prevention [13]. Despite the huge effort in the search of treatment for COVID-19, there are only 2 U.S. Food and Drug Administration (FDA) approved drugs to date. One of the current FDA-approved drugs is remdesivir, an antiviral drug targeting the RdRp of SARS-CoV-2. Remdesivir can be metabolised into an adenosine triphosphate (ATP) analog that inhibits viral RNA transcription [14]. Remdesivir has also shown activity against SARS-CoV and Middle East respiratory syndrome coronavirus (MERS-CoV) [15]. Remdesivir is approved for the treatment of COVID-19 in adults and children (≥28 days of age and weighing at least 3 kg). Patients with mild to moderate COVID-19 who are at high risk of developing severe diseases that could necessitate hospitalisation or even death can be treated with remdesivir [16].

Hospitalised patients who require mechanical ventilation, extracorporeal membrane oxygenation (ECMO), or oxygen supplementation may take an immune modulator, the Janus Kinase (JAK) 1 and 2 inhibitors, called baricitinib [16,17]. The FDA also granted baricitinib an Emergency Use Authorization (EUA) for treating paediatric patients of 2 to 18 years old who require mechanical ventilation, extracorporeal membrane oxygenation (ECMO), or oxygen supplementation. Additionally, molnupiravir targets the RdRp of SARS-CoV-2, leading to erroneous viral genome replication, and Paxlovid, which targets the main protease active site of SARS-CoV-2 by combining ritonavir and nirmatrelvir, are also approved for patients with mild-to-moderate COVID-19 [16,18]. Tocilizumab, a different interleukin-6 receptor-blocking monoclonal antibody, is approved for the treatment of COVID-19 in hospitalised adults and paediatric patients (2 years and older) receiving systemic corticosteroids [16,19]. Bebtelovimab, a monoclonal antibody that neutralised the virus and blocks virus entry into host cells, is EUA authorised for adults and paediatric patients (12 years and older and weighing at least 40 kg) with mild to moderate COVID-19 [16]. Table 1 provides a list of drugs that have received FDA and EUA approval.

### 2.2. Potential Drugs Undergoing Clinical Trials

Since the outbreak, the development of medications to combat COVID-19 has accelerated. Other medications are also recommended in addition to those that have FDA approval or FDA–EUA authorisation. The United States National Institutes of Health (NIH) established the public–private partnership Accelerating COVID-19 Therapeutic Interventions and Vaccines (ACTIV) as part of its efforts to combat COVID-19 [20]. Clinical trials for the therapeutic drugs were carried out on patients needing critical care, hospitalised/moderately ill patients, outpatients, and patients recovering from COVID-19. This effort enables efficient and accelerated drug development while eliminating those not meeting the criteria [21].

Clinical research on monoclonal antibodies, inhibitors, immunological modulators, anticoagulants, and other therapeutic drugs is now being carried out within the ACTIV alliance. Table 2 lists these medications. It is encouraging to see that most of the medications listed are now in phase 3 trials and can potentially be granted FDA approval or EUA authorisation in the near future. This review discussed the medication in phase 3 trials or later and some commonly used drugs in treating COVID-19.

Most of the monoclonal antibodies for COVID-19 belong to the immunoglobulin G1 subclass. They acted by interfering with the spike protein and ACE2 receptor binding, preventing the virus uptake [23]. Monoclonal antibodies such as bamlanivimab (LY-CoV555), AZD7442 (Evusheld), and BRII-196/BRII-198 showed promising results in clinical trials to be utlised as COVID-19 drugs [22]. In an interim analysis for bamlanivimab in a phase 2 trial, 452 patients were randomly treated with 700 mg, 2800 mg, and 7000 mg of bamlanivimab and placebo via intravenous infusion. Patients administered 2800 mg of bamlanivimab showed accelerated clearance of viral load over time [24]. A randomised phase 3 clinical trial administering 4200 mg bamlanivimab or placebo with 966 participants concluded that treatment with bamlanivimab reduced the risk of COVID-19 among the residents and staff in skilled nursing and assisted living facilities [25]. In addition, a BLAZE-1 study with 613 patients in a phase 2/3 trial aimed to study the effect of bamlanivimab monotherapy and bamlanivimab and etesevimab, another monoclonal antibody, combination therapy on viral load. Patients were administered 700 mg, 2800 mg, 7000 mg of bamlanivimab, 2800 mg of bamlanivimab, and 2800 mg etesevimab, or placebo. The results showed that patients treated with combination therapy significantly decreased viral load on the 11th day [26]. Dougan et al. also reported in a phase 3 trial involving 1035 patients with mild or moderate COVID-19 treated with 2800 mg bamlanivimab and 2800 mg etesevimab showed a decrease in hospitalisation and death due to COVID-19 compared to the placebo group [27].

AZD7442, also known as Evusheld is a combination of two long-acting monoclonal antibodies (LAABs), tixagevimab and cilgavimab, developed by AstraZeneca [28]. In the phase 3 randomised trial involving 5197 participants, 300 mg of AZD7442 and placebo were administered. The results showed that a single dose of AZD7442 administered intramuscularly is able to reduce the risk of symptomatic COVID-19. Participants accessed after 6 months in the same study showed a relatively reduced infection risk by 82.8%. Hence, a single dose of AZD7442 is effective in preventing COVID-19 and is safe for administration. This could be an alternative for people under certain medical conditions who cannot be vaccinated [29]. In another phase 3 trial involving 910 unvaccinated patients, the administration of 300 mg of AZD7442 protects against the progression of COVID-19 to a severe state or death compared to a placebo. This study also concluded that early treatment with AZD7442 in mild or moderate COVID-19 patients might result in a more favourable outcome [30].

The efficacy and safety of 1000 mg BRII-196 plus 1000 mg BRII-198, another group of monoclonal antibodies, were studied in an ACTIV-3 Therapeutics for Inpatients with COVID-19 (TICO) trial involving 546 hospitalised patients. However, the results did not show significant improvement among the patients compared to the placebo; hence, the design of the study might need to be improved [31]. In an ACTIV-2 Study for Outpatients with COVID-19 trial, 1000 mg BRII-196 and 1000 mg BRII-198 administered to COVID-19 outpatients showed efficacy, as noted in the status of the trial [22].

The progression of the COVID-19 disease results in a systemic hyperinflammatory syndrome due to the host’s response toward the viral invasion [32]. In turn, the inflammation response activates coagulation, further augmenting the inflammation response, which might result in multiple organ failure and eventually, death [33,34]. This phenomenon is characterised by an elevation of inflammatory markers and coagulation factors such as interleukin (IL)-2, IL-6, IL-7, granulocyte colony-stimulating factor, macrophage inflammatory protein 1-α, tumor necrosis factor-α (TNF-α), interferon-γ (IFN-γ) inducible protein 10, monocyte chemoattractant protein 1, C-reactive protein, ferritin, procalcitonin, D-dimer, and so on [32,35,36,37]. As a result, trials on various drugs for anticoagulant therapy and immune modulators are also involved in the ACTIV program [22,38].

Crizanlizumab, a monoclonal antibody acting on P-selectin, is administered in the hope of decreasing P-selectin levels and to suppress inflammatory and thrombotic markers. In the CRITICAL (Crizanlizumab for Treating COVID-19 Vasculopathy) trial, a single dose of 5 mg/kg crizanlizumab reduced P-selectin levels at 89%. However, the significance of this study still needs to be evaluated in a larger trial, as there were only 54 patients involved [39].

Fostamatinib, an SYK Inhibitor, can decrease inflammation, reduce immunothrombosis, and inhibit a pulmonary epithelial transmembrane protein, Mucin-1, related to the severity of acute respiratory distress syndrome (ARDS). In the phase 2 randomised trial, 150 mg fostamatinib and the placebo were administered twice daily to 59 patients. Although the sample size is small, the result showed decreased days in the intensive care unit (ICU) compared to the placebo and an increased rate of reduction in the inflammatory markers and coagulation factors. The effects are more apparent in patients in a severe or critical state. Furthermore, no deaths have occurred in patients treated with fostamatinib [40]. Fostamatinib has progressed to a phase 3 trial involving 308 hospitalised patients and a phase 3 ACTIV-4 Host Tissue trial involving 1600 patients [22,41].

Sodium/glucose cotransporter-2 (SGLT2) inhibitors are drugs used to treat diabetes. Treatment with SGLT2 inhibitors in diabetic patients showed reduced major adverse cardiovascular events by 11%, a reduction in hospitalisation due to heart failure by 31%, a composite of cardiovascular death or hospitalisation for heart failure by 23%, and a decrease in renal disease by 45%. The use of SGLT2 inhibitors also manifested a reduced mortality rate [42]. Besides, SGLT2 inhibitors reduce IL-6, and TNF-α levels which are the key inflammatory markers [43]. Since COVID-19 appears to cause similar effects, such as heart failure and renal dysfunction, it is speculated that SGLT2 can be used to improve the outcomes of COVID-19. The efficacy and safety of SGLT2 inhibitors are being evaluated in a Dapagliflozin in Respiratory Failure in Patients with COVID-19 (DARE-19) trial and in a TACTIC-E trial [43,44].

Abatacept is an immune modulator used to treat rheumatoid arthritis. Its activity in modulating T-cell activation and downregulating cytokines production while monitoring the downstream immune response has drawn interest in considering it as a drug for COVID-19 [45]. Systemic response of abatacept also showed high significant antagonism with COVID-19 severity, further supporting the use of abatacept to prevent severe COVID-19 [46]. Furthermore, a phase 3 ACTIV-1 Immune Modulators trial involving 1022 patients showed that a 10 mg/kg dose of abatacept administration reduced the mortality rate by 37.4%. Patients receiving abatacept also have enhanced clinical improvement by 34.2%. The safety profile of abatacept is also confirmed in this trial [47].

Infliximab is a monoclonal antibody that binds to TNF- α and inhibits its inflammatory effect [48]. Infliximab showed the potential to be added as a drug for COVID-19. In a phase 3 ACTIV-1 Immune Modulators trial involving 1037 patients, the mortality rate of patients receiving 5 mg/kg of infliximab decreased by 40.5%. Patients receiving infliximab also have enhanced clinical improvement by 43.8% [47,49].

An antidepressant drug, a selective serotonin reuptake inhibitor (SSRI), fluvoxamine was evaluated for drugs to combat COVID-19 due to its anti-inflammatory and antiviral effects. Fluvoxamine might lower clinical deterioration in COVID-19 patients as a σ-1 receptor (S1R) agonist which regulates cytokine production. Fluvoxamine is safe and inexpensive, and readily available in the market [50,51]. A randomised trial involving 152 unhospitalised patients administered with 100 mg fluvoxamine or placebo showed lower clinical deterioration over 15 days in the fluvoxamine group [52]. In another TOGETHER trial, 1497 high-risk, unhospitalised patients were assigned 100 mg fluvoxamine twice daily or a placebo. The results showed that patients administered fluvoxamine reduced the need for hospitalisation or emergency observation [50,53]. A systematic review and meta-analysis performed on fluvoxamine trials further affirmed that fluvoxamine reduced hospitalisation by 94.1% to 98.6% [54].

There are two types of heparin used as anticoagulants, the unfractionated heparin (UFH) and the low molecular weight heparin (LMWH) [55]. The anticoagulant properties suggested that heparin can be used to treat COVID-19. In 2021, a systematic review and meta-analysis were carried out on UHF and LMWH to evaluate their ability to reduce the mortality rate in COVID-19 patients. The analysis was performed on 33 studies involving a total of 32,688 patients. The analysis concluded that heparin, administered in full dose or prophylactic dose, is able to reduce the mortality rate in COVID-19 patients but warned that heparin administered at full dose might increase the risk of bleeding [56]. Another meta-analysis of nine studies involving 9637 patients concluded that administration with LMWH has better outcomes than UFH [57]. However, when it comes to critically ill COVID-19 patients, administering LMWH or UFH did not show an improvement in survival rate. This is reported by Volteas et al. in a study involving 218 patients [58].

Ivermectin is an anti-parasitic agent having antiviral properties. Studies on ivermectin have shown contrasting results. A meta-analysis on ivermectin involving 15 trials and 2438 patients concluded that reduction of mortality is possible with ivermectin and that early use of ivermectin may reduce the progression of COVID-19 to a more severe phase [59]. In an I-TECH randomised clinical trial involving 490 patients, patients were divided into a group that received ivermectin (0.4 mg/kg) for 5 days plus standard of care and a group that received standard of care alone. This trial concluded that early treatment of ivermectin did not prevent the progression of COVID-19 [60]. In another trial involving 3525 patients, patients were randomly assigned 400 μg/kg ivermectin, placebo, or other intervention. The result concluded that ivermectin did not prevent hospitalisation due to progression to severe disease nor reduce emergency observation time in outpatients [61]. The efficacy of ivermectin with a dose of 600 μg/kg is further analysed under the phase 3 ACTIV-6 Outpatient trial to determine its role in treating COVID-19 [22].

Although dexamethasone is not on the list of ACTIV trials due to its common use, it is also discussed in this review. Dexamethasone has anti-inflammatory properties and assists in controlling inflammatory responses. It is used for inflammatory suppression and allergic disorders. This property makes it a potential drug for treating COVID-19 and has been shown to decrease mortality in COVID-19 [62]. In the CoDEX randomised clinical trial, 299 patients received a randomly assigned dose of 20 mg of dexamethasone for 5 days, followed by 10 mg of dexamethasone for 5 days, or until ICU release. Standard care is also provided to this group, while another group receives only standard care. Results of this trial showed that the group receiving dexamethasone had a statistically significant increased number of days alive and free of mechanical ventilation over 28 days [63]. In another RECOVERY trial, 2104 patients were administered 6 mg of dexamethasone for up to 10 days, while 4321 patients received standard care. The result showed that patients administered with dexamethasone had a lower mortality rate over 28 days [64]. The COVID STEROID 2 trial aims to study whether there is a difference in number of days alive in patients with severe hypoxemia administered with 12 mg and 6 mg dexamethasone. They recruited 1000 patients in this study, in which 503 patients received 12 mg dexamethasone and 497 patients received 6 mg dexamethasone for up to 10 days. The result showed no statistically significant difference between the two groups [65].

Of the drugs listed and discussed, it is observed that most of the drugs target the responses evoked by a viral infection, such as hyperinflammation and coagulation problems, while only a few of the monoclonal antibodies target the virus itself. Hence, more antiviral drugs targeting SARS-CoV-2 are needed. There were also drugs previously granted EUA but revoked by FDA. Drugs such as REGEN-COV (casirivimab and imdevimab), sotrovimab, bamlanivimab, and etesevimab are not authorised until further notice by FDA. The reason being mutations in SARS-CoV-2 variants have rendered the drugs ineffective [16]. Hence, drugs targeting all or most SARS-CoV-2 variants are also urgently needed.

## 3. Expanding the Therapeutic Approaches for COVID-19

Despite significant efforts to produce a successful COVID-19 treatment, there are now only a small number of FDA-approved therapeutic medicines for COVID-19 therapy. Remdesivir and baricitinib are the only available treatment medications. It would be advantageous to investigate additional possibilities, such as small compounds and peptides, for developing COVID-19 medicines. The therapeutic approaches also focus on boosting the host’s immune system with monoclonal antibodies, inhibiting virus genome replication by inhibiting RdRp, and prescribing anti-inflammatory drugs and immune modulators to reduce the host’s hyper inflammation response. Other strategies, such as preventing viral entry into host cells or inhibiting the formation of RTC through inhibiting the main protease, should also be considered.

### 3.1. Peptides Targeting S Protein RBD

Several strategies prevent viral entry into host cells, including blocking RBD and ACE2 binding, preventing S protein cleavage, and blocking the binding of HR1 and HR2 by small compounds and peptides. In 2020, Cao et al. designed several mini proteins, AHB1, AHB2, LCB1, and LCB3, which bind to the RBD of the SARS-CoV-2 S protein. These proteins inhibit infection of SARS-CoV-2 to Vero E6 cells with IC_50_ of 35 nM, 15.5 nM, 23.5 pM, and 48.1 pM, respectively [66]. However, these mini proteins are a few folds larger than standard peptides, and there is no information on in vivo models. Wolfe et al. generated a list of peptides using computer modeling and further tested their binding affinity using bio-layer interferometry (BLI). They highlighted four peptides, P89, P100, P168, and P180, which bind to subunit 1 of S protein with binding constants (KD) 124 nM, 185 nM, 143 nM, and 243 nM, respectively [67]. However, there is no information on additional analysis, such as IC_50_ determination or in vivo study.

The most promising stapled peptide created by Curelli et al. to target the S protein RBD was NYBSP-4. NYBSP-4 has a KD value of 2.2 µM using surface plasmon resonance (SPR). It expressed a calculated half-life (T1/2) of >289 min in human plasma, showing high proteolytic stability. It did not demonstrate toxicity even when tested with a maximum dose and has an IC_50_ of 1.97 µM and 2.8 µM in two cell types, HT1080/ACE2 and A549/ACE2 cells, upon infection with the SARS-CoV-2 pseudovirus [68]. However, there is no investigation using in vivo models. In addition, Karoyan et al. designed three interesting peptides, P8, P9, and P10, with IC_50_ values of 46 nM, 53 nM, and 42 nM on Calu-3 cells following SARS-CoV-2 infection with KD values of 24 nM, 0.09 nM, and 0.03 nM, respectively. These peptides did not cause any cytotoxicity in Vero-E5 and Calu-3 cells [69]. However, there is no testing of these peptides in vivo. Among the list of peptides developed by Chen et al. in 2021 is a peptide called AYn1, which has a KD value of 95.6 nM using localised surface plasmon resonance (LSPR) and an IC50 value of 4.9 μM when infecting HEK293T/hACE2 with SARS-CoV-2 pseudovirus and demonstrated minimal cytotoxicity to the cells. However, the result of AYn1 in in vivo models is not satisfactory [70].

### 3.2. Peptides Targeting S Protein HR1-HR2 Fusion

Several peptides have been reported to inhibit the fusion of HR1 and HR2 of SARS-CoV-2 [71]. One peptide, EK1C4, can inhibit infection of SARS-CoV-2 pseudotyped and live virus at IC_50_ of 15.8 nM and 36.5 nM, respectively. EK1C4 is found to exhibit little or no toxic effect in vitro. The modified EK1C4 peptide, EKL1C, has an inhibitory effect on SARS-CoV-2 pseudotyped and live virus infection at IC_50_ of 3–27 nM in various cell types. EKL1C also demonstrated the ability to reduce SARS-CoV-2 virus titer in the lungs of hACE2-Tg mice. It has stronger resistance to proteolytic enzymes and exhibits higher thermostability properties. A dimeric peptide candidate, [SARSHRC-PEG4]2-chol lipopeptide, has been reported to inhibit HR1 and HR2 fusion. It inhibits viral entry after 8 h on Vero-E6 and Vero-E6-TMPRSS2 cells with IC_50_ of ~300 nM and ~5 nM, respectively, and showed no toxicity in a cellular toxicity assay [72].

### 3.3. Small Molecules and Peptides Targeting Host Proteases and SARS-CoV-2 Main Protease

Several small molecules and peptides that target host proteases have been reported. The TMPRSS2 peptide mimetic inhibitors MI-432, MI-1900, the peptide aprotinin, and the furin inhibitor MI-1851 all suppress SARS-CoV-2 replication in Calu-3 [73]. Apart from this, furin/proprotein convertase (PC) inhibitors, decanoyl-RVKR-chloromethylketone (CMK), naphthofluorescein, and a TMPRSS2 inhibitor, camostat effectively decreases SARS-CoV-2 production in Vero-E6 cells [74]. Teicoplanin, a glycopeptide antibiotic that inhibits cathepsin L, effectively halted the infection of HEK293T and Huh7 cells by SARS-CoV-2 pseudotyped viruses [75]. However, further analyses such as cytotoxicity or in vivo model tests have not been carried out in these studies.

An in vitro fluorescence resonance energy transfer (FRET) assay identified five drugs that are inhibitors of the SARS-CoV-2 main protease and have IC_50_ values of 4.81 µM, 5.4 µM, 16.2 µM, 38.5 µM, and 18.7 µM, respectively. These drugs are manidipine, boceprevir, lercanidipine, efonidipine, and bedaquiline [76]. In another study, four peptides, p12, p13, p15, and p16, showed binding to the main protease and competitively inhibiting main protease activity. The peptides have an IC_50_ value of 5.36 µM, 3.11 µM, 5.31 µM, and 3.76 µM, respectively, as determined via solid-phase extraction coupled to mass spectrometry (SPE MS) [77]. In Vero cells infected with SARS-CoV-2, two other drugs, ebselen and a Michael acceptor inhibitor known as N3, showed antiviral effects [78]. This suggests that these two drugs may be able to permeate the host cell membrane and target the primary protease. Teicoplanin, also has inhibitory activity on the SARS-CoV-2 main protease [79]. However, further analyses, such as cytotoxicity or in vivo models, are also lacking.

The peptides targeting the S protein are summarised in Table 3, while inhibitors targeting the host proteases and SARS-CoV-2 main protease are summarised in Table 4. Other studies design peptides targeting S protein in silico that are not listed here as the efficacy of these peptides remained to be determined via in vitro and in vivo analysis. A list of small molecules is currently being screened as potential therapeutic drugs. However, despite considerable effort, the development of therapeutic drugs that inhibit viral entry or disrupt the viral life cycle is still in the preclinical stage. Nevertheless, these studies showed that therapeutic drug development toward inhibiting viral entry and RTC formation is advancing. It is therefore anticipated that therapeutic medications will soon be created using this strategy.

## 4. A New Area That Can Be Considered for the Development of COVID-19 Drugs

There is an area that remained seemingly untouched in the therapeutic approach that prevents viral entry into host cells. Apart from inhibiting host cell proteases and the primary SARS-CoV-2 protease and preventing the interaction of the RBD with ACE2 and the HR1-HR2 fusion, studies on developing therapeutic drugs that can bind to the S protein’s S1/S2 cleavage site and stop host proteases from breaking it down should be considered. The S protein has an N-terminal domain (residue 14–305), RBD (residue 319–541), S1/S2 cleavage site where host proteases cleave between residue 685 and 686, fusion peptide domain (residue 788–806), HR1 (residue 912–984), HR2 (residue 1163–1213), transmembrane domain (residue 1213–1237), and cytoplasm domain (residue 1237–1273) [80]. Compared to the RBD, the S protein’s S1/S2 cleavage site is relatively conserved, as shown in Table 5 [81]. For this reason, therapeutic drugs that target S1/S2 cleavage site might face fewer challenges caused by mutation of the targeted binding site.

Developing therapeutic drugs targeting the S1/S2 cleavage site instead of host proteases seems more applicable, as the host proteases are involved in important physiological processes. Consider the PC family member furin, which is a key player in embryogenesis. Experimenting on furin knockout mice results in embryonic fatality. Furin also aids in synaptic innervation, which is another crucial role. It is accomplished by the pro-nerve growth factor’s pro-NGF neurotrophin, created by furin cleaving it and binding to the Trk receptors (NGF). Pro-NGF is secreted when furin is inhibited, and it binds to the neurotrophin receptor (p75NTR) and triggers apoptosis [82]. In addition, Cathepsin L is involved in proteolytic activities and generates active enzymes, receptors, and biologically active peptides [83].

Furthermore, furin belongs to the PC family, TMPRSS2, the transmembrane-bound serine proteases family, and cathepsin L, the cathepsin family. Proteases within a common family possess similar structures and mechanisms. Hence, the inhibitors targeting these proteases might also end up targeting proteases of the same family, disrupting important physiological functions with unforeseen consequences. As a result, targeting the S protein’s S1/S2 cleavage site might be a better course of action than suppressing the host proteases.

## 5. Conclusions

Huge efforts are still needed to develop effective therapeutic drugs to treat COVID-19. Although many therapeutic drugs are now under clinical trials, only a few FDA-approved therapeutic drugs exist. Therapeutic drugs under clinical trials are mostly immune modulators or anticoagulants. Drugs targeting SARS-CoV-2 variants are in urgent need of development. Options such as small molecules or peptides that target SARS-CoV-2 viral entrance or life cycle, as well as the host proteases, must be included in the expansion of treatment approaches and alternatives. There is a gap in these approaches, as seen in the lack of development of therapeutic drugs that target the S1/S2 cleavage site. There is a need to fill this gap for effective treatment of COVID-19 to be made available in the future, together with efforts in other strategies.

## Figures and Tables

**Table 1 pathogens-11-01201-t001:** FDA-approved and FDA–EUA-authorised drugs [16].

Type	Drug	Treatment Target
FDA approved	Remdesivir	Adults and children (≥28 days of age and weighing at least 3 kg), hospitalised or not hospitalised, who have mild to moderate COVID-19 and are at high risk of developing severe diseases that could necessitate hospitalisation or even death
	Baricitinib	Hospitalised adults who require mechanical ventilation, extracorporeal membrane oxygenation (ECMO), or oxygen supplementation
FDA–EUA authorised	Baricitinib	Paediatric patients of 2 to 18 years old who require mechanical ventilation, extracorporeal membrane oxygenation (ECMO), or oxygen supplementation
	Molnupiravir	Adults with mild to moderate COVID-19
	Ritonavir and nirmatrelvir	Adults and paediatric patients (12 years and older and weighing at least 40 kg) with mild to moderate COVID-19
	Tocilizumab	Hospitalised adults and paediatric patients (2 years and older) receiving systemic corticosteroids
	Bebtelovimab	Adults and paediatric patients (12 years and older and weighing at least 40 kg) with mild to moderate COVID-19

**Table 2 pathogens-11-01201-t002:** Therapeutic drugs in ACTIV partnership under clinical trials [22].

Type	Drug	Clinical Trial
Monoclonal Antibody	SkyrisiTM (risankizumab) for inpatient.	Phase 2
	Lenzilumab for inpatient.	Phase 2
	LY-CoV555 for outpatient.	Phase 2/3
	AZD7442 for outpatient.	Phase 2/3
	BRII-196 and BRII-198 for outpatient.	Phase 2/3
	AZD7442 for inpatient.	Phase 3
	Crizanlizumab for inpatient.	Phase 4
Factor D Inhibitor	Danicopan for inpatient.	Phase 2
Spleen Tyrosine Kinase (SYK) Inhibitor	Fostamatinib.	Phase 3
Sodium/glucose cotransporter-2 inhibitor	SGLT2i.	Phase 4
Immune Modulators	Orencia^®^ (abatacept) for inpatients.	Phase 3
	Remicade^®^ (infliximab) for inpatients.	Phase 3
Immune Modulators/Antiviral	Fluvoxamine for outpatient.	Phase 3
	Ivermectin 600 mcg for outpatient.	Phase 3
Anticoagulants	Unfractionated (UF) and Low Molecular Weight (LMW) heparin for inpatients.	Phase 4
Others	Convalescent Plasma for inpatient.	Phase 2
	Veklury^®^ (remdesivir) for inpatient.	Phase 3
	Veklury^®^ (remdesivir) and Olumiant^®^ (baricitinib) for inpatients.	Phase 3

**Table 3 pathogens-11-01201-t003:** Peptides targeting SARS-CoV-2.

Peptides	Peptide Sequence	K_D_	IC_50_	Toxicity	In vivo Models	Reference
**Peptides are designed to target the RBD of S protein.**						
AHB1	DEDLEELERLYRKAEEVAKEAKDASRRGDDERAKEQMERAMRLFDQVFELAQELQEKQTDGNRQKATHLDKAVKEAADELYQRVRELEEQVMHVLDQVSELAHELLHKLTGEELERAAYFNWWATEMMLELIKSDDEREIREIEEEARRILEHLEELARK	-	35 nM	-	-	[66]
AHB2	ELEEQVMHVLDQVSELAHELLHKLTGEELERAAYFNWWATEMMLELIKSDDEREIREIEEEARRILEHLEELARK	-	15.5 nM	-	-	
LCB1	DKEWILQKIYEIMRLLDELGHAEASMRVSDLIYEFMKKGDERLLEEAERLLEEVER	-	23.54 pM	-	-	
LCB3	NDDELHMLMTDLVYEALHFAKDEEIKKRVFQLFELADKAYKNNDRQKLEKVVEELKELLERLLS	-	48.1 pM	-	-	
P89	DWTLFLFVFNLEWEDLFY	124 nM	-	-	-	[67]
P100	KTEWDKWMHMYYEIFYED	185 nM	-	-	-	
P168	NFDILLFVFNYEMEDKFY	143 nM	-	-	-	
P180	KTDWDMFSHWMEIYFYVI	243 nM	-	-	-	
NYBSP-4 *	-	2.2 µM	1.97–2.8 µM	No	-	[68]
P8	SALEEQLKTFLDKFMHELEDLLYQLAL	24 nM	46 nM	No	-	[69]
P9	SALEEQYKTFLDKFMHELEDLLYQLSL	0.09 nM	53 nM	No	-	
P10	SALEEQYKTFLDKFMHELEDLLYQLAL	0.03 nM	42 nM	No	-	
AYn1	KKKKKKDKFNHEAEDLFY	95.6 nM	4.9 µM	No	Not satisfactory	[70]
**Peptides are designed to target HR1 and HR2 of SARS-CoV-2.**						
EK1C4	SLDQINVTFLDLEYEMKKLEEAIKKLEESYIDLKEL-GSGSG-PEG4-C(chol)	-	36.5 nM	No	-	[71]
EKL1C	NVTFLDLEYEMKKLEEAIKKLEESYIDLKELGTVEY-GSG-C(Chol)	-	3 nM	No	Reduce SARS-CoV-2 titer	
[SARSHRC-PEG4]2-chol	[DISGINASWNIQKEIDRLNEVAKNLNESLIDLQEL-PEG4]2-chol	-	~300 nM and ~5 nM	No	-	[72]

* Sequence for NYBSP-4 not shown as it is a double-stapled peptide.

**Table 4 pathogens-11-01201-t004:** Small molecules or peptides as inhibitors of host cell protease and SARS-CoV-2 main protease.

Inhibitors	Inhibitory Effect	Reference
**TMPRSS2 inhibitors:**		
MI-432	Suppresses replication of SARS-CoV-2 in Calu-3 cells	[73]
MI-1900		
Aprotinin		
Camostat	Decreases SARS-CoV-2 production in Vero-E6 cells	[74]
**Furin inhibitors:**		
MI-1851	Suppresses replication of SARS-CoV-2 in Calu-3 cells	[73]
CMK	Decreases SARS-CoV-2 production in Vero-E6 cells	[74]
Naphthofluorescein		
**Cathepsin L inhibitor:**		
Teicoplanin	Inhibits SARS-CoV-2 pseudotyped virus entry in HEK293T and Huh7 cells	[75]
**SARS-CoV-2 main protease inhibitors:**		
Manidipine	IC_50_ = 4.81 µM	[76]
Boceprevir	IC_50_ = 5.4 µM	
Lercanidipine	IC_50_ = 16.2 µM	
Efonidipine	IC_50_ = 38.5 µM	
Bedaquiline	IC_50_ = 18.7 µM	
p12	IC_50_ = 5.36 µM	[77]
p13	IC_50_ = 3.11 µM	
p15	IC_50_ = 5.31 µM	
p16	IC_50_ = 3.76 µM	
Ebselen	Antiviral effects in Vero cells infected with SARS-CoV-2	[78]
N3		
Teicoplanin	IC_50_ = 1.5 µM	[79]

**Table 5 pathogens-11-01201-t005:** Mutations in the variants of SARS-CoV-2 [81].

SARS-CoV-2 Variant	Mutation on RBD	Mutation near S1/S2 Cleavage Site	Mutation on Other Sites
Alpha	N501Y	P681H	D614G
Beta	K417N, E484K, N501Y	-	D614G, A701V
Gamma	K417T, E484K, N501Y	-	D614G, H655Y
Delta	L452R, T478K	P681R	D614G
Omicron	G339D, S371L, S373P, S375F, K417N, N440K, G446S, S477N, T478K, E484A, Q493R, G496S, Q498R, N501Y, Y505H	P681H	A67V, Δ69-70, T95I, G142D, Δ143-145, N211I, Δ212, ins215EPE, T547K, D614G, H655Y, N679K, N764K, D796Y, N856K, Q954H, N969K, L981F

## Data Availability

Not applicable.

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
