# Peer review of "COVID-19 Treatment—Current Status, Advances, and Gap"

_pathogens, 2022, doi:10.3390/pathogens11101201_

Round 1

Reviewer 1 Report

This manuscript by Ho and Lee entitled "COVID-19 Treatment - Current Status, Advances, and Gap" gives a summary of the current research for clinical management of COVID-19 however the manuscript needs further improvement before the recommendation for publication. 

The reviewer comments are:

1. There are already reviews that show the approaches, gaps, and treatment for COVID-19, what is novel about your study? Add this part to your discussion and conclusions 

2. Aside from the S1/S2 Cleavage site as a target for drug/vaccine development, what are the other gaps that need to be addressed? 

3. For tables 1-4, All the summary tables should have references cited from which study this information was lifted. 

4. For current approaches, it would be appreciated if there is a table summary of the current available/approaches for the treatment of COVID-19 used worldwide, including all the FDA-approved drugs for use. 

5. Also, please improve the presentation of the table for ease of readers' understanding. 

Reviewer 2 Report

Comments in .pdf file.

Reviewer 3 Report

A review article by  Chian Ho and Ping-Chin Lee is devoted to an important topic. It summarizes information on various strategies designed to block the SARS-CoV-2. It is interesting that the authors tried to talk about scientific works that are poorly represented in other reviews.

At the same time, I have a few comments for the authors.

1. In Table 1, it is better in the line of inhibitors, add inhibitors of what these drugs? In general, monoclonal antibodies are formally also inhibitors, they are inhibitors of the entry of the virus into the cell.

2. Line 130, error in concentration, this is too high a concentration (most likely not M but μM).

3. Lines 165-166, error in concentration, this is too high a concentration (most likely not M but nM).

Round 2

Reviewer 1 Report

the manuscript (pathogens-1906310) entitled ‘COVID-19 Treatment – Current Status, Advances, and Gap’ discusses the current advances for Covid-19. The reviewer commends the authors for the improvement of the manuscript. The reviewer believes that the manuscript may improve greatly if the authors would send this for English editing for clarity of understanding. Overall, the manuscript can be of great help to give updates and insights to science researchers and medical field workers.

Reviewer 2 Report

All my requests were appropriately answered. This new version of the manuscript has been highly improved.

I suggest this new manuscript version should be evaluated by an additional reviewer who has not seen the previous version to determine if this article is publishable in Pathogens.